rsos.royalsocietypublishing.org

synthetic chemistry/organic chemistry

synthesis, 15N-labelled, purine, cytokinin

**Author for correspondence:**
Jan Buček
e-mail: bucek89@gmail.com

This article has been edited by the Royal Society of Chemistry, including the commissioning, peer review process and editorial aspects up to the point of acceptance.

# Total synthesis of [15N]-labelled C6-substituted purines from [15N]-formamide—easy preparation of isotopically labelled cytokinins and derivatives

Jan Buček[1], Marek Zatloukal[1], Libor Havlíček[2],
Lucie Plíhalová[1,3], Tomáš Pospíšil[1], Ondřej Novák[3],
Karel Doležal[1,3] and Miroslav Strnad[3]

[1]Department of Chemical Biology and Genetics, Centre of the Region Haná for Biotechnological and Agricultural Research, Faculty of Science, Palacký University, 78371 Olomouc-Holice, Czech Republic
[2]Isotope Laboratory, Institute of Experimental Botany ASCR, Vídeňská 1083, 14220 Prague, Czech Republic
[3]Laboratory of Growth Regulators, Centre of the Region Haná for Biotechnological and Agricultural Research, Faculty of Science, Palacký University and Institute of Experimental Botany, Academy of Sciences of the Czech Republic, 78371 Olomouc-Holice, Czech Republic

KD, 0000-0003-4938-0350

Cytokinins (CKs) and their metabolites and derivatives are essential for cell division, plant growth regulation and development. They are typically found at minute concentrations in plant tissues containing very complicated biological matrices. Therefore, defined standards labelled with stable isotopes are required for precise metabolic profiling and quantification of CKs, as well as *in vivo* elucidation of CK biosynthesis in various plant species. In this work, 11 [15N]-labelled C6-purine derivatives were prepared, among them 5 aromatic (**4, 5, 6, 7, 8**) and 3 isoprenoid (**9, 10, 11**) CKs. Compared to current methods, optimized syntheses of 6-amino-9H-[15N5]-purine (adenine) and 6-chloro-9H-[15N4]-purine (6-chloropurine) were performed to achieve more effective, selective and generally easier approaches. The chemical identity and purity of prepared compounds were confirmed by physico-chemical analyses (TLC; HRMS; HPLC–MS; 1H, 13C, 15N NMR). The presented approach is applicable for the synthesis of any other desired [15N4]-labelled C6-substituted purine derivatives.

rsos.royalsocietypublishing.org   R. Soc. open sci. **5**: 181322

# 1. Introduction

Cytokinins (CKs) are naturally occurring substances in plants derived from adenine with either an aromatic (ARCK) or isoprenoid (ISCK) side chain at the $N^6$-position. CKs can regulate all stages of plant development and directly affect plant growth processes [1].

Both types of naturally occurring CKs were discovered, quantified and characterized [2–6]. Subsequently, newly prepared CKs and their derivatives were synthesized and tested in several studies for various biological activities [7,8]. Some of the derivatives exhibit a strong cytotoxic effect on human cancer cell lines [9,10]. Others contribute to important applications in pharmacology and cosmetics [11]. As apparent from the number of publications dedicated to various biological effects of CKs and their derivatives, development of sensitive and robust analytical methods for monitoring endogenous concentrations of CKs in various biological systems became crucial.

Analysing CKs in complex biological matrices is difficult because they exist in very low concentrations (pmol g$^{-1}$ fresh weight) [12]. Modern analytical procedures for the determination of CKs consist of sample pre-treatment and subsequent instrumental measurement of individual CK metabolites [13]. At present, endogenous CK metabolites are generally quantified by mass spectrometry (MS) using the isotope dilution technique [12]. Isotopically labelled standards are easily distinguished during MS analysis due to their unique masses. The stable isotope dilution method, which involves determining the concentration of a non-labelled (endogenous) compound and comparing it to that of a labelled internal standard, can be very accurate and precise. Therefore, isotopically labelled CK standards are highly beneficial for controlling selectivity, affinity, recovery and capacity of newly developed analytical procedures and for correction of ion suppression effects during MS analysis.

Preparation methods for CK standards containing naturally occurring nuclides are well-established, documented and functional [7,14,15]. On the contrary, for isotopically labelled CKs, existing preparation methods are problematic and need optimization. Several approaches were recently developed. Hydrogen and carbon nuclides ($^2$H, $^3$H, $^{13}$C) are most widely used in the fields of phytohormones and plant physiology. However, considering the characteristics of the molecules prepared herein, only preparation methods for CKs labelled with stable isotopes will be described.

Among ARCKs, the $^2$H and $^{15}$N isotopologues of 6-benzylaminopurine (BAP) have already been prepared. Deuterium can be incorporated with a catalysed hydrogen–deuterium exchange reaction [16]. A reaction between 6-chloropurine and the corresponding isotopically labelled [$^{15}$N$_1$] amine can be used to achieve a more stable [$^{15}$N$_1$]-BAP [17,18]. However, the above-mentioned methods can only produce the [$^{15}$N$_1$] isotopomer, which is insufficient for use as an internal standard for MS.

Considerable efforts were made to prepare an isotopically labelled analogue of *trans*-zeatin (*t*Z) since it was one of the first discovered and frequently occurring ISCKs. However, chemical, spectroscopic and enzymatic evidence suggest that the zeatin molecule exists as *cis*- and *trans*-geometrical isomers [3]. Thus, a stereospecific synthetic approach had to be developed first to produce a particular geometric isomer. Synthesis of the *trans*-isomer was reported in 1966, based on the reaction of the crude amine with 6-methylthiopurine [19]. However, the first reliable stereospecific method for preparing [$^{13}$C$_8$]-*t*Z was presented 5 years later [20]. Alternatively, $^2$H nuclide can be introduced to the *t*Z side chain as well to produce [$^2$H$_5$]-*t*Z [21]. A few years later, alongside the first attempt to prepare $^{15}$N$_4$ *trans*-zeatin [22], another complex approach has been developed, which gradually combines side chain catalytic constructions [23]. By using deuterated reduction systems and solvents, isotopically labelled *t*Z side chains can be produced for reaction with 6-chloropurine [23].

Currently published approaches for isotopically labelled isopentenyladenine (iP) preparation are based mainly on side chain isotopic labelling. Introduction of $^{13}$C nuclide into the iP structure can proceed via de novo side chain construction using $^{13}$CO$_2$ as the nuclide donor [24]. Catalytic reduction of a specific nitrile to an amine using LiAD$_4$ with a subsequent reaction of the amine with 6-chloropurine produced twice-deuterated iP at the $N^6$ position [25].

Isotopically labelled CK standards contributed in part to several recent studies focused on: plant physiology [26,27]; CK profiling [12,13,27,28]; interspecies interactions [29,30]; and, finally, CK biosynthesis and metabolism [31,32]. Interestingly, deuterated CK standards were used in most of the above-mentioned studies.

Although several approaches to producing isotopically labelled CK standards have been published, no functional and complete approach for easy and relatively inexpensive production from a simple precursor exists. Most of the above-mentioned studies used hydrogen nuclides. Without accounting for radioactive $^3$H due to its inherent dangers and lack of comfort with its use, even $^2$H is a poor

choice for metabolomic applications because its physico-chemical properties are vastly different from the naturally occurring hydrogen isotope [33]. The possibility of hydrogen–deuterium exchange during laboratory and *in vivo* manipulations present another important disadvantage. These and other considerations as well as the purine-core structure make use of the $^{15}$N nuclide a reasonable choice. Despite the existence of previous studies describing [$^{15}$N$_5$] adenine and *trans*-zeatin preparation [22,34] as well as formamide cyclization to adenine [35], a significant improvement of continuous and fully functional synthesis of $^{15}$N-core-labelled CKs from [$^{15}$N]-formamide is presented herein. Furthermore, the yields are strongly improved compared to the less effective procedures described above.

# 2. Material and methods

## 2.1. General procedures

### 2.1.1. Equipment

Analytical thin-layer chromatography (TLC) was performed using silica gel ALUGRAM Xtra SIL G/ UV$_{254}$ plates (Macherey-Nagel, Düren, Germany). An ultraviolet (UV) cabinet with adjustable UV lengths 254/364 nm (Camag, Muttenz, Switzerland) was used for detection. The melting points were determined on Büchi Melting Point B-540 apparatus and are uncorrected. High-performance liquid chromatography–UV–diode array–mass spectrometry (HPLC–UV–DAD–MS) experiments were performed using a Waters 2695 separation module linked with a Waters 2996 photodiode array detector (PDA; Waters, Milford, MA, USA), followed by a hybrid quadrupole time-of-flight (Q-TOF) Micro$^{TM}$ mass spectrometer equipped with electrospray ionization interface (Waters MS Technologies, Manchester, UK). High-resolution mass spectrometry (HRMS) was used to determine the elemental composition of prepared compounds. HRMS involved an ultra-performance liquid chromatography– UV–diode array–mass spectrometry (HPLC–UV–DAD/HPLC–MS) experiment using an Acquity UPLC H-Class system (Waters, Milford, MA, USA) followed by a hybrid Q-TOF tandem mass spectrometer Synapt G2-Si equipped with electrospray ionization interface (Waters MS Technologies, Manchester, UK). Data were processed using MassLynx 4.1 software. NMR spectra were measured by a JEOL ECA-500 spectrometer operating at 19°C and 500.16 MHz ($^1$H), 125.77 MHz ($^{13}$C) and 50.68 MHz ($^{15}$N), respectively. Samples were prepared by dissolving the compounds in DMSO-d6. Tetramethylsilane (TMS) for $^1$H and $^{13}$C, and ($^{15}$N)-ammonium for $^{15}$N, were used as external standards.

### 2.1.2. HPLC–UV–DAD and HPLC–MS conditions

Compounds (1 mg) were dissolved in 1 ml of 1% methanol and injected (10 µl) onto a reversed-phase column (Symmetry C18, 5 µm, 150 × 2.1 mm; Waters, Milford, MA, USA) incubated at 25°C. Solvent A was 15 mM ammonium formate adjusted to pH 4.0. Solvent B was methanol. The following binary gradient was used at a flow-rate of 200 µl min$^{-1}$: 0 min, 10% B; 0–24 min linear gradient to 90% B; 25–34 min isocratic elution of 90% B; 35–45 min linear gradient to 10% B. The flow was introduced to a DAD detector (scanning range 210–400 nm with 1.2 nm resolution) and then to an electrospray source (source temperature 120°C, desolvation temperature 300°C, capillary voltage 3 kV, cone voltage 20 V). Nitrogen was used as the cone gas (50 l h$^{-1}$) and the desolvation gas (500 l h$^{-1}$). Data acquisition was performed in full-scan mode (50–1000 Da) with a scan time of 0.5 s and a collision energy of 6 eV; argon was used as the collision gas (optimized pressure of $5 \times 10^{-3}$ mbar). Analyses were performed in positive mode (ESI$^+$), therefore protonated molecules [M+H]$^+$ were collected in each MS spectrum. HPLC–UV purity was determined for every prepared compound. The percentage result was calculated as the representation of the molecular peak area compared to the sum of the remaining peak areas in the entire HPLC–UV spectrum.

### 2.1.3. HRMS conditions

Samples were prepared as described above (*HPLC–MS conditions*). Samples (5 µl) were injected onto a reversed-phase column (Symmetry C18, 5 µm, 150 mm × 2.1 mm; Waters, Milford, MA, USA) incubated at 40°C. Solvent A was 15 mM ammonium formate adjusted to pH 4.0. Solvent B was methanol. The following linear gradient was used at a flow rate of 250 µl min$^{-1}$: 0 min, 10% B; 0– 15 min, 90% B. The effluent was introduced to a DAD detector (scanning range 210–400 nm with 1.2 nm resolution) and then to an electrospray source (source temperature 150°C, desolvation

temperature 550°C, capillary voltage 1 kV, cone voltage 25 V). Nitrogen was used as the cone gas ($50 \, l \, h^{-1}$) and the desolvation gas ($1000 \, l \, h^{-1}$). Data acquisition was performed in full-scan mode (50–1000 Da) with a scan time of 0.5 s and collision energy of 4 eV; argon was used as the collision gas (optimized pressure of $5 \times 10^{-3}$ mbar). Analyses were performed in positive mode (ESI$^+$), therefore protonated molecules [M+H]$^+$ were collected in each MS spectrum. For the exact mass determination experiments, the external calibration was performed using lock spray technology and a mixture of leucine/encephalin (50 pg $\mu l^{-1}$) in an acetonitrile and water (1 : 1) solution with 0.1% formic acid as a reference. Accurate masses were calculated and used to determine the elemental composition of the analytes with a fidelity better than 1.0 ppm.

### 2.1.4. Calculating isotopologue abundance

The isotopologue composition of each prepared compound was calculated as described below. HRMS analysis was performed to obtain the most accurate spectra. Subsequently, every isotopologue was identified and its presence in proportion to the whole mixture was calculated. For instance, the MS spectra in figure 1 demonstrate isotopologue enumeration for 6-amino-9H-[$^{15}$N$_5$]-purine. However, an isotopologue abundance calculation method was subsequently applied to each synthesized compound.

## 2.2. Chemicals

[$^{15}$N]-formamide (99.1% $^{15}$N enrichment based on starting materials) was obtained from Cambridge Isotope Laboratories (Andover, USA). N,N-diisopropylethylamine (DIPEA), tert-butyl methyl ether (MTBE), 3-methoxybenzylamine, benzylamine, DMSO-d6 and Dowex 50 W were obtained from Sigma-Aldrich. Phosphorus oxychloride (POCl$_3$) was obtained from Merck Millipore. Lachner supplied n-propanol and acetic acid (CH$_3$COOH). Penta supplied ammonium hydroxide solution (NH$_4$OH), sodium nitrite (NaNO$_2$), triethylamine, methanol and chloroform. Olchemim Ltd (Olomouc, Czech Republic) supplied 3-methylbut-2-en-1-amine hydrochloride, 4-amino-2-methylbut-2-en-1-ol hemitrate salt, 2-hydroxy-, 3-hydroxy- and 4-hydroxybenzylamines. Milli-Q water was used throughout. The other solvents and chemicals used were all of standard p.a. quality.

## 2.3. Synthesis

### 2.3.1. 6-amino-9H-[$^{15}$N$_5$]-purine (adenine) (1)

[$^{15}$N$_5$]-adenine was prepared by cyclization of [$^{15}$N]-formamide in the presence of POCl$_3$, as previously described [35]. A reaction mixture containing [$^{15}$N]-formamide (6.0 ml; 0.15 M) and POCl$_3$ (28.2 ml; 0.30 M) was placed into a stainless-steel reactor with a polytetrafluoroethylene (PTFE) tube insert and stirred for 17 h at 130°C under argon. After cooling, the mixture was transferred to a flask containing Dowex 50 W (H$^+$ form, 150 g) and water (200 ml). The contents were extensively washed with water and the product was eluted using NH$_4$OH (5 M) after 2 days of stirring, as described [22]. Next, the solution was evaporated to constant weight. Owing to the already-described occurrence of the stereoisomer 1H-imidazo-[4,5-b]pyrazine-5-amine [36], the reaction mixture was finally purified by flash chromatography on a silica gel column using chloroform/methanol/ammonia (6 : 1 : 0.05) as the mobile phase. Collected fractions were evaporated again to give the final product. Yield: 1.5 g white crystal (32%). TLC (chloroform/methanol/ammonia 6 : 1 : 0.05, v/v/v): one single spot, free of 1H-imidazo-[4,5-b]pyrazine-5-amine. Melting point 372–375°C. HRMS (ESI$^+$): m/z 141.0475 [M+H]$^+$ (Calcd for [C$_5$H$_5^{15}$N$_5$+H]$^+$ 141.0475). MS (ESI$^+$): m/z 140.89 [M+H]$^+$. HPLC–UV purity: 98+%. $^{15}$N$_5$ isotopologue abundance: 95.0%. $^1$H NMR (500 MHz, DMSO-$d_6$) δ ppm 6.98 (br. s., 1 H, HN-6) 7.15 (br. s., 1 H, HN-6) 8.00–8.11 (m, 2 H, H-2, H-8) 12.81 (br. s., 1 H, HN-9). $^{13}$C NMR (126 MHz, DMSO-$d_6$) δ ppm 119.1 (C-5), 139.5 (C-8), 150.6 (C-4), 152.8 (C-2), 156.1 (C-6). $^{15}$N NMR (51 MHz, DMSO-$d_6$) δ ppm 75.3 (d, J = 4.6 Hz, 1 N, N-6) 153.3 (s, 1 N, N-9) 223.8 (s, 1 N, N-3) 230.3 (d, J = 5.2 Hz, 1 N, N-1) 236.4 (s, 1 N, N-7).

### 2.3.2. 1,7-dihydro-6H-[$^{15}$N$_4$]-purine-6-one (hypoxanthine) (2)

[$^{15}$N$_4$]-hypoxanthine was prepared by [$^{15}$N$_5$]-adenine deamination in a weak acidic medium containing nitric salts, as previously described [37]. Sodium nitrite (4.5 g; 0.06 M) was added to the suspension of [$^{15}$N$_5$]-adenine (1.5 g; 0.01 M) and acetic acid (30%; 41 ml; 0.2 M), and the mixture was heated to 40°C

rsos.royalsocietypublishing.org    R. Soc. open sci. **5**: 181322

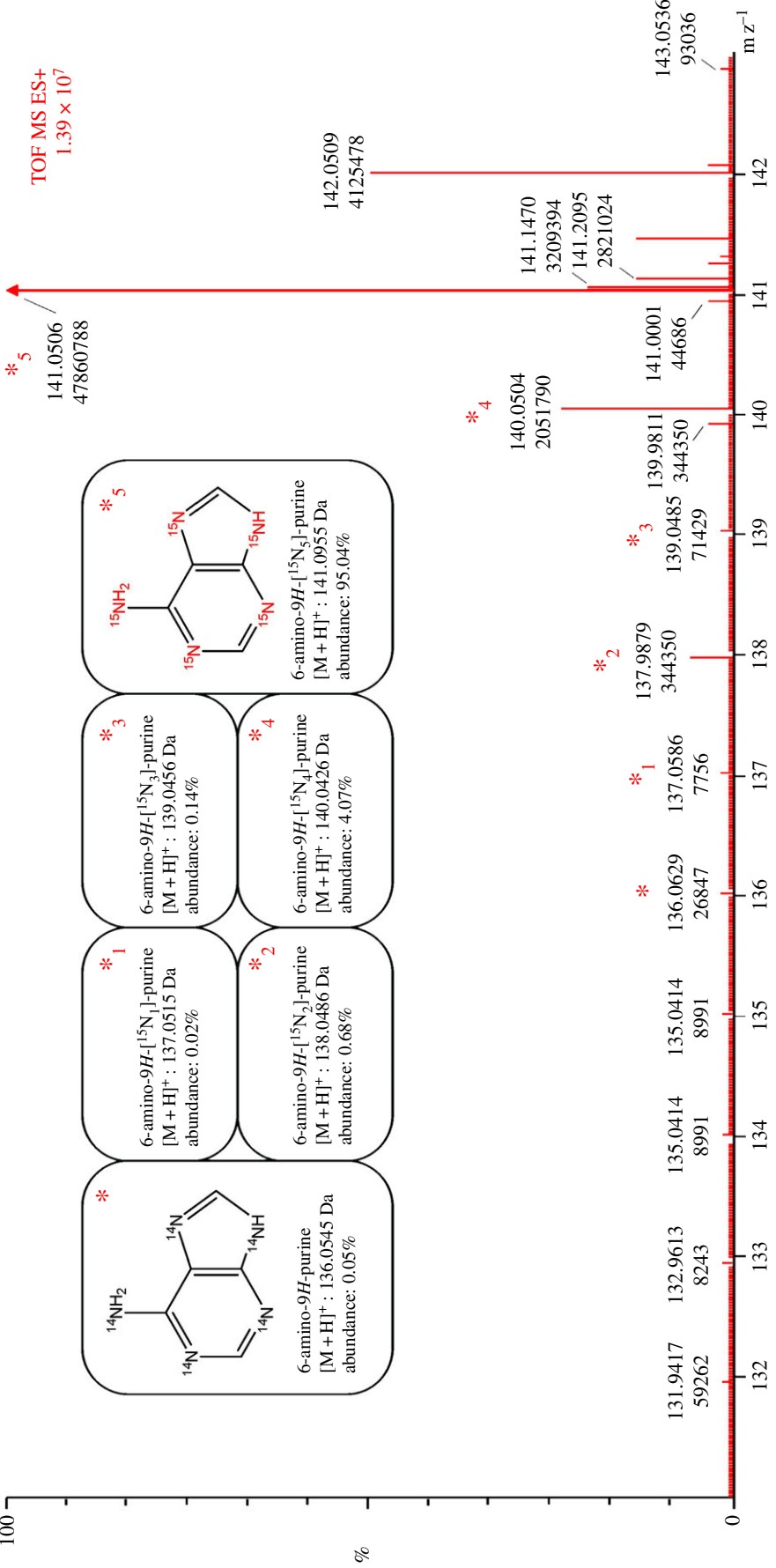

**Figure 1.** HRMS spectra of 6-amino-9H-[$^{15}$N$_5$]-purine and its respective isotopologues. Theoretical monoisotopic weights of protonated form for identifying molecular ion peak and calculated isotopologue abundance are mentioned.

and stirred for 1 h. Next, the reaction mixture was left to crystallize at 4°C. After crystallization, the precipitate was filtered off, washed with water (3 × 5 ml) and dried at 60°C to constant weight. Yield: 1.1 g white crystal (71%). TLC (chloroform/methanol/ammonia 6:1:0.05, v/v/v): one single spot, free of starting compound. HRMS (ESI$^+$): $m/z$ 141.0345 [M+H]$^+$ (Calcd for [C$_5$H$_4$$^{15}$N$_4$O+H]$^+$ 141.0345). MS (ESI$^+$): $m/z$ 141.06 [M+H]$^+$. HPLC–UV purity: 97+%. $^1$H NMR (500 MHz, DMSO-d6) δ ppm 7.81– 8.00 (m, 1 H, H-2) 8.06 (br. s., 1 H, H-8) 12.09 (br. s., 1 H) 12.27 (br. s., 1 H). $^{13}$C NMR (126 MHz, DMSO-$d_6$) δ ppm 115.8 (C-5), 139.1 (C-8), 142.4 (C-2), 144.8 (C-6), 157.5 (C-4).

### 2.3.3. 6-chloro-9H-[$^{15}$N$_4$]-purine (6-chloropurine) (3)

[$^{15}$N$_4$]-6-chloropurine preparation was based on chlorination of [$^{15}$N$_4$]-hypoxanthine, as previously described [38]. DIPEA (3.1 ml; 0.02 M) was slowly added to a mixture of [$^{15}$N$_4$]-hypoxanthine (1.1 g; 7.8 mM) and POCl$_3$ (44 ml; 0.47 M), and the reaction was stirred under reflux for 4 h at 130°C. POCl$_3$ was then removed by distillation under reduced pressure. The POCl$_3$-free solution was subsequently transferred into a flask containing 15 ml of MTBE and an equal volume of cold water while stirring for 30 min. The water phase was removed and then subjected to extraction using MTBE again a total of 10 times (10 × 15 ml). Organic fractions collected were evaporated to constant weight. Yield: 1.2 g slightly yellow crystal (95%). TLC (chloroform/methanol/ammonia 6 : 1 : 0.05, v/v/v): one single spot, free of starting compound. Melting point 177°C. HRMS (ESI$^+$): $m/z$ 159.0006 [M+H]$^+$ (Calcd for [C$_5$H$_3$$^{15}$N$_4$Cl+H]$^+$ 159.0006). MS (ESI$^+$): $m/z$ 158.81 [M+H]$^+$. HPLC–UV purity: 97+ %. $^1$H NMR (500 MHz, DMSO-$d_6$) δ ppm 8.63 (s, 1 H, H-8) 8.68 (s, 1 H, H-2) 12.23 (br. s., 1H, HN-9). $^{13}$C NMR (126 MHz, DMSO-$d_6$) δ ppm 129.7 (C-5), 146.7 (C-8), 148.2 (C-6), 152.0 (C-2), 154.5 (C-4). $^{15}$N NMR (51 MHz, DMSO-$d_6$) δ ppm 173.15 (s, 1 N, N-9) 224.4 (s, 1 N, N-7) 251.90 (s, 1 N, N-3) 268.78 (s, 1 N, N-1).

## 2.4. Synthesis of [$^{15}$N$_4$] aromatic cytokinins

General procedures for preparing non-labelled monohydroxylated 6-benzylaminopurines (4, 5, 6) have previously been described [18]. Preparation methods for 6-(3-methoxybenzylamino)-9H-purine (7) and 6-benzylamino-9H-purine (8) have also been previously published [6,39]. In general, preparation of corresponding substituted ARCKs was based on [$^{15}$N$_4$]-6-chloropurine (100 mg) reaction with the appropriate amine and triethylamine (molar ratio 1 : 1 : 2) in cold (0°C) $n$-propanol at 100°C for 5 h in an inert atmosphere (Ar). After cooling, the reaction mixture was left to crystallize at room temperature for 24 h. Then, the precipitate was filtered out, washed with $n$-propanol (3 × 5 ml) and water (3 × 5 ml) and dried at 60°C to constant weight. Yields and analytical data (TLC; HRMS; HPLC–MS; $^1$H, $^{13}$C NMR) are mentioned in the relevant section below.

### 2.4.1. 6-(4-hydroxybenzylamino)-9H-[$^{15}$N$_4$]-purine (para-topolin, pT) (4)

Yield: 124.1 mg white crystal (80%). TLC (ethyl acetate/methanol/ammonia 34 : 4 : 2, v/v/v): one single spot, free of starting compound. HRMS (ESI$^+$): $m/z$ 246.0924 [M+H]$^+$ (Calcd for [C$_{12}$H$_{11}$N$^{15}$N$_4$O+H]$^+$ 246.0923). MS (ESI$^+$): $m/z$ 246.11 [M+H]$^+$. HPLC–UV purity: 98+ %. $^1$H NMR (500 MHz, DMSO-d6) δ ppm 4.52 (br. s., 2 H) 6.62 (m, $J$ = 8.41 Hz, 2 H) 7.11 (m, $J$ = 8.41 Hz, 2 H) 7.91–8.08 (m, 2 H) 8.13 (t, $J$ = 15.21 Hz, 1 H) 9.22 (br. s., 1 H) 12.62 (br. s., 1 H). $^{13}$C NMR (126 MHz, DMSO-d6) δ ppm 42.95, 115.43, 119.13, 129.15, 130.82, 139.37, 149.85, 152.83, 154.60, 156.63.

### 2.4.2. 6-(2-hydroxybenzylamino)-9H-[$^{15}$N$_4$]-purine (ortho-topolin, oT) (5)

Yield: 120.9 mg white crystal (78%). TLC (ethyl acetate/methanol/ammonia 34 : 4 : 2, v/v/v): one single spot, free of starting compound. HRMS (ESI$^+$): $m/z$ 246.0923 [M+H]$^+$ (Calcd for [C$_{12}$H$_{11}$N$^{15}$N$_4$O+H]$^+$ 246.0923). MS (ESI$^+$): $m/z$ 245.99 [M+H]$^+$. HPLC–UV purity: 99+ %. $^1$H NMR (500 MHz, DMSO-d6) δ ppm 4.53 (br. s., 2 H) 6.67 (t, $J$ = 7.37 Hz, 1 H) 6.75 (d, $J$ = 7.95 Hz, 1 H) 7.01 (t, $J$ = 7.49 Hz, 1 H) 7.09 (d, $J$ = 7.34 Hz, 1 H) 7.99–8.12 (m, 2 H) 8.12–8.20 (m, 1 H) 10.09 (br. s., 1 H) 12.78 (br. s., 1 H). $^{13}$C NMR (126 MHz, DMSO-d6) δ ppm 116.07, 119.44, 126.27, 128.48, 129.22, 139.69, 149.90, 152.52, 154.44, 155.57. $^{15}$N NMR (51 MHz, DMSO-d6) δ ppm 153.8, 220.6, 223.4, 235.7.

### 2.4.3. 6-(3-hydroxybenzylamino)-9H-[$^{15}$N$_4$]-purine (meta-topolin, mT) (6)

Yield: 85.4 mg white crystal (55%). TLC (ethyl acetate/methanol/ammonia 34 : 4 : 2, v/v/v): one single spot, free of starting compound. HRMS (ESI$^+$): $m/z$ 246.0925 [M+H]$^+$ (Calcd for [C$_{12}$H$_{11}$N$^{15}$N$_4$O+H]$^+$

246.0923). MS (ESI$^+$): $m/z$ 246.09 [M+H]$^+$. HPLC–UV purity: 98+ %. $^1$H NMR (500 MHz, DMSO-d6) δ ppm 3.62 (br. s., 1 H) 4.53 (br. s., 1 H) 6.49 (br. s., 1 H) 6.65 (br. s., 2 H) 6.97 (d, $J$ = 7.26 Hz, 1 H) 8.06 (br. s., 2 H) 9.19 (br. s., 1 H) 12.84 (br. s., 1 H). $^{13}$C NMR (126 MHz, DMSO-d6) δ ppm 43.15, 113.98, 114.37, 118.23, 129.64, 139.28, 139.35, 142.25, 150.06, 152.86, 154.73, 157.78.

### 2.4.4. 6-(3-methoxybenzylamino)-9$H$-[$^{15}$N$_4$]-purine (meta-methoxytopolin, memT) (7)

Yield: 139.4 mg white crystal (85%). TLC (ethyl acetate/methanol/ammonia 34 : 4 : 2, v/v/v): one single spot, free of starting compound. HRMS (ESI$^+$): $m/z$ 260.1080 [M+H]$^+$ (Calcd for [C$_{13}$H$_{13}$N$^{15}$N$_4$O+H]$^+$ 260.1080). MS (ESI$^+$): $m/z$ 260.07 [M+H]$^+$. HPLC–UV purity: 99+ %. $^1$H NMR (500 MHz, DMSO-d6) δ ppm 3.65 (s, 3 H, O-CH$_3$) 4.62 (br. s., 2 H, N-CH$_2$) 6.72 (d, $J$ = 7.1 Hz, 1 H, C$_{Ar}$) 6.84–6.89 (m, 2 H, C$_{Ar}$) 7.15 (t, $J$ = 7.8 Hz, 1 H, C$_{Ar}$) 8.02–8.09 (m, 1 H, H-8) 8.13 (d, $J$ = 15.9 Hz, 1 H, H-2) 12.91–12.99 (br. m, 1H, HN-9). $^{13}$C NMR (126 MHz, DMSO-d6) δ ppm 43.27, 55.41, 112.27, 113.47, 119.84, 129.75, 139.32, 139.40, 142.41, 149.98, 152.83, 154.74, 159.71.

### 2.4.5. 6-benzylamino-9$H$-[$^{15}$N$_4$]-purine (BAP) (8)

Yield: 59.5 mg white crystal (41%). TLC (ethyl acetate/methanol/ammonia 34 : 4 : 2, v/v/v): one single spot, free of starting compound. HRMS (ESI$^+$): $m/z$ 230.0984 [M+H]$^+$ (Calcd for [C$_{12}$H$_{11}$N$^{15}$N$_4$O+H]$^+$ 230.0974). MS (ESI$^+$): $m/z$ 230.00 [M+H]$^+$. HPLC–UV purity: 99+ %. $^1$H NMR (500 MHz, DMSO-$d_6$) δ ppm 4.65 (br. s., 2 H, N-CH$_2$) 7.11–7.19 (m, 1 H, $p$-H$_{Ar}$) 7.23 (t, $J$ = 7.5 Hz, 2 H, $o$-H$_{Ar}$) 7.29 (d, $J$ = 7.34 Hz, 2 H, $m$-H$_{Ar}$) 8.01–8.09 (m, 1 H, H-2) 8.09–8.26 (m, 2 H, H-8, HN-6) 12.81–12.99 (br. m, 1 H, HN-9). $^{13}$C NMR (126 MHz, DMSO-$d_6$) δ ppm 43.3 (NH-CH$_2$), 127.0 (C$_{Ar}$), 127.6 (C$_{Ar}$), 128.7 (C$_{Ar}$), 139.3 (C-8), 140.7 (C-4), 150.0 (C$_{Ar}$), 152.8 (C-2), 154.7 (C-6).

## 2.5. Synthesis of [$^{15}$N$_4$] isoprenoid cytokinins

The preparation of 6-[(3-methylbut-2-en-1-yl)]-9$H$-purine-6-amine (9) has previously been described [7] and its isotopically labelled analogue was prepared in a similar manner. [$^{15}$N$_4$]-6-chloropurine (50 mg; 0.3 mM) was dissolved in $n$-propanol (830 µl; 11 mM), and (3-methylbut-2-en-1-yl)amine hydrochloride (39 mg; 0.4 mM) was added in the presence of triethylamine (181 µl; 1.3 mM). The reaction was performed at 100°C for 5 h in an inert atmosphere (Ar). After cooling to room temperature, crystallization was immediately observed in the reaction mixture. The resulting crystal was filtered out, washed with $n$-propanol (3 × 2 ml) and water (3 × 2 ml) and dried at 60°C to constant weight. Melting point 201–204°C.

The syntheses of 6-[($E$)-4-hydroxy-3-methylbut-2-en-1-yl]-9$H$-[$^{15}$N$_4$]-purine-6-amine (10) and 6-[($Z$)-4-hydroxy-3-methylbut-2-en-1-yl]-9$H$-[$^{15}$N$_4$]-purine-6-amine (11) were based on the original protocols [15] with slight modifications. These syntheses were generally based on the reaction of [$^{15}$N$_4$]-6-chloropurine (50 mg) with the appropriate amine in the presence of DIPEA (molar ratio 1 : 2 : 4) and excess methanol at 85°C for 48 h in an inert atmosphere (Ar) in a pressure tube. The reaction solvents were then evaporated and replaced with water. The product was then crystallized from water at reduced temperature for 48 h. Yields and analytical data (TLC; HRMS; HPLC–MS; $^1$H, $^{13}$C NMR) are mentioned in the relevant section below.

### 2.5.1. 6-[($E$)-4-hydroxy-3-methylbut-2-en-1-yl]-9$H$-[$^{15}$N$_4$]-purine-6-amine (trans-zeatin, tZ) (10)

Yield: 28.2 mg white crystal (40%). TLC (chloroform/methanol 86 : 14, v/v): one single spot, free of starting compound. Melting point 198–200°C. HRMS (ESI+): $m/z$ 224.1082 [M+H]$^+$ (Calcd for [C$_{10}$H$_{13}$N$^{15}$N$_4$O+H] + 224.1080). MS (ESI+): $m/z$ 224.10 [M+H]$^+$. HPLC–UV purity: 97+ %. $^1$H NMR (500 MHz, DMSO-d6) δ ppm 1.61 (s, 3 H, CH$_3$) 3.73 (br. s., 2 H, CH$_2$-O) 4.05 (br. s., 2 H, NH-C$\underline{H_2}$) 4.70 (br. s., 1 H, -OH) 5.47 (br. s., 1 H, CH=) 7.66 (br. s., 1 H, $\underline{H}$N-CH$_2$) 7.96–8.06 (m, 1 H, H-8) 8.11 (t, $J$ = 14.90 Hz, 1 H, H-2) 12.93 (br. s., 1 H, HN-9). $^{13}$C NMR (126 MHz, DMSO-$d_6$) δ ppm 14.1 (CH$_3$), 37.7 (N-CH$_2$), 66.3 (CH$_2$-O), 119.2 (C-5), 121.4 (CH=), 137.6 (C=), 139.0 (C-8), 149.9 (C-4), 152.8 (C-2), 154.6 (C-6).

### 2.5.2. 6-[($Z$)-4-hydroxy-3-methylbut-2-en-1-yl]-9$H$-[$^{15}$N$_4$]-purine-6-amine (cis-zeatin, cZ) (11)

Yield: 29.0 mg white crystal (41%). TLC (chloroform/methanol 86 : 14, v/v): one single spot, free of starting compound. HRMS (ESI+): $m/z$ 224.1080 [M+H]$^+$ (Calcd for [C$_{10}$H$_{13}$N$^{15}$N$_4$O+H] + 224.1080).

rsos.royalsocietypublishing.org    R. Soc. open sci. 5: 181322

MS (ESI+): $m/z$ 224.00 [M+H]$^+$. HPLC–UV purity: 96+ %. $^1$H NMR (500 MHz, DMSO-d6) $\delta$ ppm 1.64 (s, 3 H) 3.98 (s, 2 H) 4.06 (br. s., 2 H) 4.72 (br. s., 1 H) 5.29 (br. s., 1 H) 7.59 (br. s., 1 H) 7.98–8.21 (m, 2 H) 12.77–12.94 (br. s., 1 H). $^{13}$C NMR (126 MHz, DMSO-d6) $\delta$ ppm 21.64, 37.32, 55.31, 60.16, 123.80, 137.98, 139.09, 149.75, 152.69, 154.41.

# 3. Results and discussion

Eleven [$^{15}$N$_4$]-core-labelled purine derivatives were prepared in this work, from which six have not been prepared previously. Excluding the starting compounds and intermediates (**1**, **2**, **3**), five aromatic (**4**, **5**, **6**, **7**, **8**) and three (**9**, **10**, **11**) isoprenoid cytokinins were synthesized. The identity and purity of each prepared compound were verified by TLC, HPLC–UV–DAD/HPLC–MS, HRMS, and $^1$H, $^{13}$C and $^{15}$N NMR. Melting point data, measured for some of the prepared compounds, were in very good agreement with previously published data for their unlabelled counterparts, although a little bit lower [40,41], probably due to isotopic effect. $^{15}$N NMR analyses were performed only for highly important intermediates (**1**, **3**) and one aromatic cytokinin, ortho-topolin (**5**), as an example of the end-product. The $^{15}$N NMR shifts obtained for **1** and **3** are in good agreement with literature data [34,42]. When CH$_3^{15}$NO$_2$ was used in literature for shift calibration, then this shift was corrected by 380.5 ppm for comparison with the $^{15}$N shift obtained with calibration on liquid $^{15}$NH$_3$. The abundance of major isotopologues is mentioned in details also only for parent molecule (**1**) to confirm the presumed transfer of the $^{15}$N nuclide from commercially available [$^{15}$N]-formamide. Isotopologic profiles of the remaining prepared compounds confirmed theoretical expectations. The abundance of the non-labelled forms of each prepared compound was less than 0.2% (data not shown). Synthesis of some compounds in their non-labelled form had previously been described [6,7,15,17,35,37,38]. However, most synthesis procedures were optimized significantly to synthesize labelled compounds on minimalistic scales. Thus, the number of purifications and other post-reaction steps were reduced in effort to produce reasonable yields. Scheme 1 summarizes synthesis work flows and detailed reaction conditions. Table 1 presents the yields obtained, HPLC–UV purities and the results of HPLC–MS and HRMS analyses.

The preparation of pure, fully labelled parent molecule **1** was critical for later synthesis. The reaction between formamide and POCl$_3$ in a 1 : 2 molar ratio and under defined conditions should give product (**1**) [35]. However, after HPLC–UV, mass spectrometry and $^1$H NMR analyses (data not shown), a contaminant, 1$H$-imidazo-[4,5-b]-pyrazine-5-amine (**1b**) was discovered in a 1 : 1 ratio with adenine (**1**). Authors [35] did not observe the formation of this contaminant. A study published 10 years later using the same reaction conditions confirmed the production of **1b** but also presented a methodology for its elimination based on stereospecific transformation of **1** to its picrate salts. Using this improved method, 93% of **1** contained in the reaction mixture should be isolated [36]. Unfortunately, after several repetitions, we were unable to obtain similar yield. Based on the results of HPLC–MS analyses (data not shown), **1** was not fully isolated. However, whether the method by [13] applies at the milligram scale used for our approach is questionable.

Based on this consideration, we decided to focus on developing a more effective, single-stage separation method while maintaining the starting conditions described by [35]. Reactions were performed in a stainless-steel reactor with a PTFE tube insert, which led to increased reaction stability and homogeneity. Subsequently, POCl$_3$ was removed by distillation. The reaction mixture was subjected to ion exchange chromatography performed by Dowex 50 W (H+ form) for 48 h due to the elimination of unreacted intermediates formed during supposed multistep cyclization of 1 (scheme 2) [43]. Elimination of these unreacted intermediates is crucial for the flawless course of later steps. Next, **1** and **1b** were eluted using 5 M NH$_4$OH as described elsewhere [22]. Finally, the reaction mixture containing **1** (65.8%) and **1b** (31.6%) was subjected to column chromatography using chloroform/methanol/ammonia (6 : 1 : 0.05) as the mobile phase. The final product (**1**) was isolated with a total yield of 32% and HPLC–UV purity of 98%.

Fractions containing **1b** were collected and subjected to HPLC–UV–DAD/HPLC–MS analysis to confirm chemical identity. Figure 2 shows HPLC chromatograms of **1** and **1b** isolation. Considering some of the previously published procedures for preparing **1** from simple precursors, relatively low yields are typical for this approach, due to multistep cyclization with several intermediates [22,35,36]. However, for further preparations of isotopically labelled CKs, high chemical purity and isotopic enrichment are especially crucial.

Since isotopically labelled CKs were the target products of our synthesis approach, **3** played a crucial role as the acceptor of aromatic or isoprenoid side chains during nucleophilic substitution at the

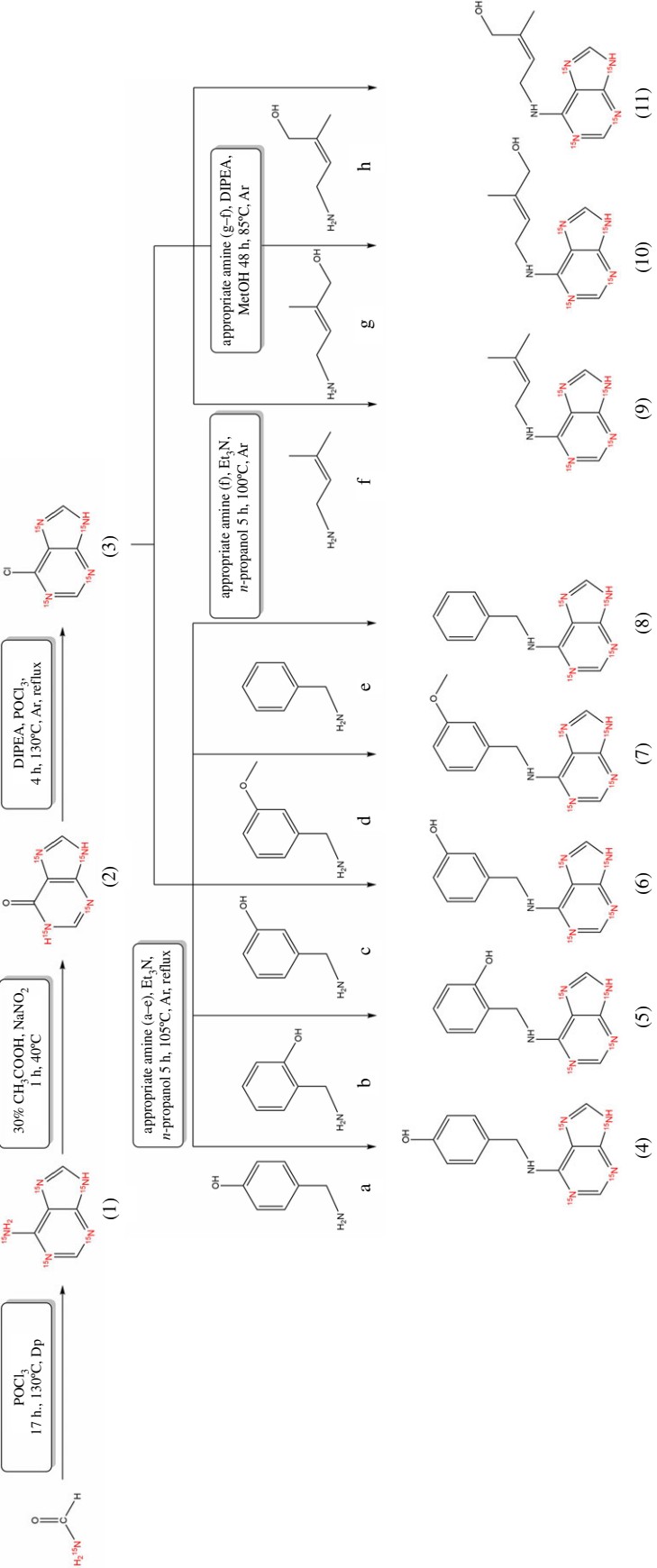

**Scheme 1.** Reaction scheme of the preparation of [$^{15}N_4$]-labelled purine derivatives. Detailed reaction conditions are mentioned in the brackets. (*a*) 4-hydroxybenzylamine; (*b*) 2-hydroxybenzylamine; (*c*) 3-hydroxybenzylamine; (*d*) 3-methoxybenzylamine; (*e*) benzylamine; (*f*) 3-methylbut-2-en-1-amine; (*g*) (E)-4-amino-2-methylbut-2-en-1-ol; (*h*) (Z)-4-amino-2-methyl-2-buten-1-ol.

**Scheme 2.** Supposed multistep 6-amino-9H-[$^{15}N_5$]-purine (**1**) cyclization from [$^{15}N$]-formamide. [$^{15}N_5$]1H-imidazo-(4,5-b)-pyrazine-5-amine (**1b**) shown in brackets because its cyclization pathway is supposedly similar to that of **1** but is not further investigated herein [43].

**Table 1.** Yields and results of physico-chemical analyses performed for each of the prepared compounds.

| compound | yield (%) | HPLC purity (%) | MS [M+H]$^+$ | high resolution mass spectrometry | | molecular formula | fidelity (ppm) |
| | | | | measured mass [M+H]$^+$ | calculated mass [M+H]$^+$ | | |
|---|---|---|---|---|---|---|---|
| **1** | 32 | 98[a] | 140.89 | 141.0475 | 141.0475 | $C_5H_5^{15}N_5$ | 0.0 |
| **2** | 71 | 97[b] | 141.06 | 141.0345 | 141.0345 | $C_5H_4^{15}N_4O$ | 0.0 |
| **3** | 95 | 97[c] | 158.81 | 159.0006 | 159.0006 | $C_5H_3^{15}N_4Cl$ | 0.0 |
| **4** | 80 | 98[b] | 246.11 | 246.0924 | 246.0923 | $C_{12}H_{11}N^{15}N_4O$ | 0.4 |
| **5** | 78 | 99[b] | 245.99 | 246.0923 | 246.0923 | $C_{12}H_{11}N^{15}N_4O$ | 0.0 |
| **6** | 55 | 98[b] | 246.09 | 246.0925 | 246.0923 | $C_{12}H_{11}N^{15}N_4O$ | 0.8 |
| **7** | 85 | 99[b] | 260.07 | 260.1080 | 260.1080 | $C_{13}H_{13}N^{15}N_4O$ | 0.7 |
| **8** | 41 | 99[b] | 230.00 | 230.0984 | 230.0974 | $C_{12}H_{11}N^{15}N_4O$ | 0.4 |
| **9** | 40 | 99[b] | 207.90 | 208.1132 | 208.1131 | $C_{10}H_{13}N^{15}N_4$ | 0.5 |
| **10** | 40 | 97[d] | 224.10 | 224.1082 | 224.1080 | $C_{10}H_{13}N^{15}N_4O$ | 0.9 |
| **11** | 41 | 96[d] | 224.00 | 224.1080 | 224.1080 | $C_{10}H_{13}N^{15}N_4O$ | 0.0 |

[a]Purified by column chromatography.
[b]Crystallization from reaction mixture.
[c]Purified by *tert*-butyl methyl ether: $H_2O$ extraction.
[d]Purified by crystallization from $H_2O$.

$C^6$-position. For this reason, we made significant efforts to obtain **3** in maximal yield and purity, and simultaneously tried to reduce the number of post-reaction steps. The preparation of **3** is generally based on hypoxanthine chlorination in the presence of the appropriate base and chlorine donor [38]. While phosphorus oxychloride ($POCl_3$) is typically the chlorine donor, choosing a suitable base is a matter of optimization. Approaches using *N,N*-dimethylaniline (DMA) [38,44], as well as those in which the reaction is conducted without the base, are known [45].

rsos.royalsocietypublishing.org    R. Soc. open sci. **5**: 181322

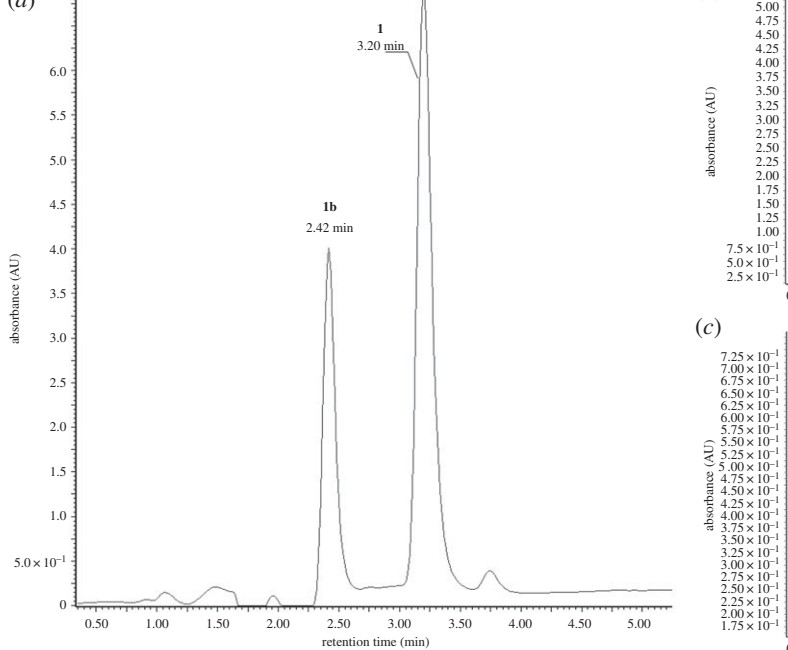

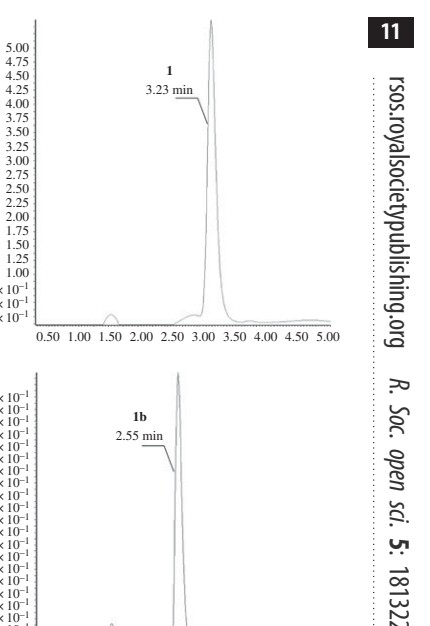

**Figure 2.** HPLC–UV chromatograms of a non-purified mixture (*a*) of [$^{15}$N$_5$]-adenine (1) and [$^{15}$N$_5$]-1H-imidazo-[4,5-b]-pyrazine-5-amine (1b). After application of column chromatography separation using chloroform/methanol/ammonia (6 : 1 : 0.05) as the mobile phase, fully separated 1 and 1b were observed as evident from *b* and *c*, respectively.

Owing to the apparent simplicity and relatively good yields (53%) of the base-less approach, we tried it several times. However, our attempts at this approach did not lead to the desired product. Therefore, we increased the reaction temperature from 65°C to 100°C. This temperature increase produced **3** with sufficient HPLC–UV purity (97%) but less than 10% yield.

Subsequent attempts to synthesize **3** were based on the approach in [22,38] but with some modifications. DMA was at first replaced with triethylamine (TEA), the temperature was increased from room temperature to 105°C, the reaction time was reduced from 24 h to 4 h, and finally, the reaction was followed by discontinuous extraction in an ethylacetate : H$_2$O system (1 : 1). These attempts produced **3** with a 14% yield, and HPLC–UV purity of 85% with no further purifications. Reaction conditions were further modified to increase the yield, so a 48 h long continuous extraction using diethyl ether was introduced. As expected, the yield increased up to 26%, while HPLC–UV purity remained below 90%.

Considering the results of previous optimization attempts, the duration of the whole method (72 h), and instrumental difficulties (continuous extraction, several pH adjustments, etc.), we shifted our attention to developing more efficient, more reliable and simpler methods. Our main goals were to select a functional base and to avoid time-consuming product extraction. The chlorine donor, POCl$_3$, was preserved, and reaction conditions, i.e. a temperature of 130°C and a duration of 4 h, were established. We tried 2,6-dimethylpyridine as a base at first, but with no success. Owing to its wide application in organic chemistry, *N,N*-diisopropylethylamine (DIPEA, Hünigs base) was finally used, instead of TEA. Selection of hypoxanthine : POCl$_3$ : DIPEA (1 : 6 : 2) at defined reaction conditions, together with subsequent discontinuous product extraction by *tert*-butyl methyl ether (M*T*BE) : H$_2$O system (1 : 1) led to the desired production of **3** with 95% yield and HPLC–UV purity of 97%. This simple and very efficient method for preparing **3** with a reaction time shorter than 8 h is presented herein. Table 2 summarizes the results of optimization, including the partial results according to the reaction conditions.

The final step in the synthesis of aromatic and isoprenoid CKs was conducted as previously published in the literature without any major changes. The preparation of **8** was performed either with a non-substituted ring [39] or accompanied by various monohydroxy- (**4**, **5**, **6**) or monomethoxy- (**7**) substituents at the phenyl ring [6,39]. Isoprenoid CKs were prepared using previously published procedures (**9**) or with slight modifications in which TEA was replaced with DIPEA and the reaction time was extended up to 48 h (**10**, **11**) [7,15]. Since any significant modifications to the preparation methods were made to the last reaction step (C$^6$-conjugation), the yields and purities of all newly

**Table 2.** Results of the optimization process of 1,7-dihydro-6H-($^{15}N_4$)-purine-6-one (hypoxanthine, 1) chlorination to 6-chloro-9H-($^{15}N_4$)-purine (6-chloropurine, 2).

| reactants[a] | reaction conditions | | additional purification step | yield (%) | HPLC purity (%) |
|---|---|---|---|---|---|
| | temperature (°C) | time (h) | | | |
| acetonitrile, ethylbenzene | 65 | 6 | a | n.d. | n.d. |
| acetonitrile, ethylbenzene | 65–80 | 6 | a | n.d. | n.d. |
| acetonitrile, ethylbenzene | 100 | 6 | a | <10 | 97 |
| triethylamine | 105 | 4 | b | 14 | 85 |
| triethylamine | 105 | 4 | c | 25 | <90 |
| 2,6-dimethylpyridine | 130 | 4 | a | n.d. | n.d. |
| N,N-diisopropylethylamine | 130 | 4 | d | 95 | 97 |

[a]Including hypoxanthine and POCl$_3$ in all cases.

a, none; b, discontinuous extraction by ethylacetate : H$_2$O (1 : 1); c, continuous extraction (48 h) by diethyl ether; d, discontinuous extraction by tert-butyl methyl ether : H$_2$O (1 : 1).

synthesized compounds were comparable to those mentioned in the literature [7,15]. The consequences of isotopic labelling with $^2$H, $^{13}$C and especially $^{15}$N, used in this work, will be further discussed below.

Generally, CKs labelled with deuterium or $^{13}$C nuclide are most frequently prepared [16,20,21,23–25]. For $^{15}$N labelling, to the best of our knowledge, for aromatic cytokinins only the preparation of [$^{15}$N$_1$]-BAP has been described in the literature [17,18]. The only [$^{15}$N$_4$]-labelled cytokinin prepared previously was trans-zeatin and its riboside, synthesized by Horgan & Scott [22]. Their basic strategy was similar to ours; however, with very low yield and unreported isotopic purity of the final products. The prevalence of approaches that use deuterium or $^{13}$C nuclide to prepare isotopically labelled CKs is reasonable. Deuterium labelling is relatively inexpensive and can be accomplished using deuterated catalytic reduction systems [23,25] or catalysed hydrogen-deuterium exchange [16]. The $^{13}$C nuclide could easily be integrated by using $^{13}$C reagents [20,23,24].

However, use of $^2$H or $^{13}$C nuclides has its disadvantages. First, most of the preparation methods mentioned are based on incorporation of one [20,23,24] or approximately two [20,23] isotopes. Truthfully, there are some preparation methods for CKs labelled with multiple isotopes. [$^2$H$_5$]-tZ can be obtained by using [$^2$H$_6$]-acetone as a starting compound for the preparation of (E)-4-amino-1,1-[$^2$H$_2$]-2-[$^2$H$_3$]-methylbut-2-en-1-ol, which is subsequently reacted with 6-chloropurine to get the desired ISCK. Additionally, [$^2$H$_4$]-BAP can be obtained by a hydrogen–deuterium exchange reaction catalysed by palladium on a carbon-ethylene diamine complex [Pd/C(en)]. In this method, the hydrogen–deuterium exchange occurs most frequently at C2, C8 (94%) and side chain carbon positions (97%) [16].

It is noteworthy that the preparation methods described are based on integration of approximately two nuclides into the purine core. If there are more isotopes in the structure, they are typically located on the side chain. Side chain locations of isotopes could be problematic for further biological application because, while a purine core is relatively stable during metabolism, side chains are often being transformed. Therefore, for biological application, the weights of compounds prepared with side chain isotopes are displaced by an approximate two-unit weight shift. This shift casts doubt on the reliability of these compounds for use as internal standards in MS [12].

Moreover, use of $^2$H nuclides can introduce further complications due to their different physico-chemical properties compared to naturally occurring nuclides. Specifically, the literature explains that some deuterated forms of drugs demonstrate different transport processes, increased resistance to metabolic change, or even changes to the entire pathway of its metabolism [33]. Moreover, use of

rsos.royalsocietypublishing.org   R. Soc. open sci. 5: 181322

deuterated internal standards can cause unstable retention times associated with the number of deuterium atoms used. The so-called deuterium isotope effect can lead to the worst accuracy and precision for a quantification method [46].

The method presented herein provides full purine-core-labelled CKs with a stable $^{15}$N nuclide that is free of side effects and that produces target compounds with high yield efficiency and high chemical purities. High abundance of the most enriched isotopologue, with the non-labelled form present at under 0.05% abundance, is further guaranteed by $^{15}$N nuclide stability as well as full purine-core cyclization at the beginning of the whole approach using simple [$^{15}$N]-formamide as the starting compound. Thus, space for incorporation of natural nitrogen isotopes is minimized.

Since every prepared adenine derivative contains at least four $^{15}$N atoms at the stable purine-core positions, they are fully useful for MS applications [47]. Moreover, this preparation method is applicable for the preparation of any other desired [$^{15}$N]-labelled C6-substituted purine derivatives. Although the applications of some compounds discussed or presented in this paper have already been published [47], a preparation method for these compounds is presented here for the very first time.

# 4. Conclusion

In summary, 11 [$^{15}$N$_4$]-core-labelled purine derivatives were synthesized, including five ARCKs (**4**, **5**, **6**, **7** and **8**) and three ISCKs (**9**, **10** and **11**). Effective modifications of previously published procedures led to enhanced selectivity of product preparation for **1** and more effective overall synthesis of **3**. According to the results of the analyses performed, the identity of all 11 compounds was confirmed and their purity proved sufficient for further applications. Moreover, the approaches presented are applicable for synthesizing any other desired [$^{15}$N]-labelled C6-substituted purine derivatives.

Data accessibility. This article has no additional data: all the data and other materials required to allow a reader to perform a full replication are available in the main body of the manuscript.

Authors' contributions. M.Z., L.H., K.D. and M.S. conceived the research idea and designed the experiments. J.B., M.Z., L.H. and L.P. participated on synthesis; T.P. and O.N. measured and analysed NMR and HR-MS data, respectively. J.B. wrote the manuscript with help of KD. All the authors read, edited and approved the final version of the manuscript.

Competing interests. We declare we have no competing interests.

Funding. Financial support was provided by project GA16-04184S from the Czech Science Foundation, as well as by the Ministry of Education, Youth and Sports, Czech Republic (grant no. LO1204 from the National Program of Sustainability I. as well as European Regional Development Fund-Project 'Centre for Experimental Plant Biology': no. CZ.02.1.01/0.0/0.0/16_019/0000738)

Acknowledgements. The authors gratefully acknowledge Jana Kočířová for her technical help.

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
