## [Reviewer comments · Royal Society Open Science]

Review History

RSOS-181322.R0 (Original submission)

Review form: Reviewer 1 (Qi-Huang Zheng)

Is the manuscript scientifically sound in its present form?

Yes

Are the interpretations and conclusions justified by the results?

Yes

Is the language acceptable?

Yes

Is it clear how to access all supporting data?

Not Applicable

Do you have any ethical concerns with this paper?

No

Have you any concerns about statistical analyses in this paper?

No

Recommendation?

Accept with minor revision (please list in comments)

Comments to the Author(s)

RSOS-181322

This paper describes total synthesis of [15N]-labelled C6-substituted purines from [15N]-formamide – easy preparation of isotopically-labelled cytokinins and derivatives. Significant efforts have been devoted to the project, and some new chemistry data have been reported. The results are reasonable and support the conclusions. It fits well in the scope of RSOS, and it will be suitable for publication after careful revision.

Comments:

1. References section is listed in alphabetical order, thus the references cited in the text is in a very unusual format, not starting from 1. Is this format RSOS requires?
2. Too many references are cited in this manuscript, it might decrease the novelty of the work, making it an improvement of the synthetic approach.
3. References 71 and 72 seem not in alphabetical order. Please double check.
4. For solid compounds, it is better to give mp data.
5. For ¹H NMR data, it is better to give the proton assignments.
6. HRMS, it is noted that the calcd value is exactly same with found value for several compounds. Please double check.
7. Within 11 compounds prepared, clearly point out how many new compounds and known compounds in Results and Discussion section.
8. Page 14, Scheme 2 caption, there are some format issues: Italic, Superscript, Subscript; and compound number should be Bold.

Review form: Reviewer 2

Is the manuscript scientifically sound in its present form?

Yes

Are the interpretations and conclusions justified by the results?

Yes

Is the language acceptable?

Yes

Is it clear how to access all supporting data?

Yes

Do you have any ethical concerns with this paper?

No

Have you any concerns about statistical analyses in this paper?

No

Recommendation?

Accept as is

Comments to the Author(s)

The paper describes a practical and efficient synthesis of a useful labelled building block that should find use in the isotope labelling community.

Decision letter (RSOS-181322.R0)

18-Sep-2018

Dear Dr Dolezal:

Title: Total synthesis of [15N]-labelled C6-substituted purines from [15N]-formamide – easy preparation of isotopically-labelled cytokinins and derivatives
Manuscript ID: RSOS-181322

Thank you for submitting the above manuscript to Royal Society Open Science. On behalf of the Editors and the Royal Society of Chemistry, I am pleased to inform you that your manuscript will be accepted for publication in Royal Society Open Science subject to minor revision in accordance with the referee suggestions. Please find the reviewers' comments at the end of this email.

The reviewers and handling editors have recommended publication, but also suggest some minor revisions to your manuscript. Therefore, I invite you to respond to the comments and revise your manuscript.

Please also include the following statements alongside the other end statements. As we cannot publish your manuscript without these end statements included, if you feel that a given heading is not relevant to your paper, please nevertheless include the heading and explicitly state that it is not relevant to your work. We have included a screenshot example of the end statements for reference.

- Ethics statement

Please clarify whether you received ethical approval from a local ethics committee to carry out your study. If so please include details of this, including the name of the committee that gave consent in a Research Ethics section after your main text. Please also clarify whether you received informed consent for the participants to participate in the study and state this in your Research Ethics section.

OR

Please clarify whether you obtained the necessary licences and approvals from your institutional animal ethics committee before conducting your research. Please provide details of these licences and approvals in an Animal Ethics section after your main text.

OR

Please clarify whether you obtained the appropriate permissions and licences to conduct the fieldwork detailed in your study. Please provide details of these in your methods section.

- Acknowledgements

Because the schedule for publication is very tight, it is a condition of publication that you submit the revised version of your manuscript before 27-Sep-2018. Please note that the revision deadline will expire at 00.00am on this date. If you do not think you will be able to meet this date please let me know immediately.

Best wishes,
Dr Laura Smith, MRSC

Publishing Editor, Journals
Royal Society of Chemistry,
Thomas Graham House,
Science Park, Milton Road,
Cambridge, CB4 0WF, UK

Royal Society Open Science - Chemistry Editorial Office

On behalf of the Subject Editor Professor Anthony Stace and the Associate Editor Professor John Moses.

RSC Associate Editor:
Comments to the Author:
(There are no comments.)

RSC Subject Editor:
Comments to the Author:
(There are no comments.)

Reviewer comments to Author:
Reviewer: 1

Comments to the Author(s)
RSOS-181322

This paper describes total synthesis of [15N]-labelled C6-substituted purines from [15N]-formamide – easy preparation of isotopically-labelled cytokinins and derivatives. Significant efforts have been devoted to the project, and some new chemistry data have been reported. The results are reasonable and support the conclusions. It fits well in the scope of RSOS, and it will be suitable for publication after careful revision.

Comments:

1. References section is listed in alphabetical order, thus the references cited in the text is in a very unusual format, not starting from 1. Is this format RSOS requires?
2. Too many references are cited in this manuscript, it might decrease the novelty of the work, making it an improvement of the synthetic approach.
3. References 71 and 72 seem not in alphabetical order. Please double check.
4. For solid compounds, it is better to give mp data.
5. For ¹H NMR data, it is better to give the proton assignments.
6. HRMS, it is noted that the calcd value is exactly same with found value for several compounds. Please double check.
7. Within 11 compounds prepared, clearly point out how many new compounds and known compounds in Results and Discussion section.
8. Page 14, Scheme 2 caption, there are some format issues: Italic, Superscript, Subscript; and compound number should be Bold.

Reviewer: 2

Comments to the Author(s)

The paper describes a practical and efficient synthesis of a useful labelled building block that should find use in the isotope labelling community.

Author's Response to Decision Letter for (RSOS-181322.R0)

See Appendix A.

Decision letter (RSOS-181322.R1)

16-Oct-2018

Dear Dr Dolezal:

Title: Total synthesis of [15N]-labelled C6-substituted purines from [15N]-formamide – easy preparation of isotopically-labelled cytokinins and derivatives

Manuscript ID: RSOS-181322.R1

It is a pleasure to accept your manuscript in its current form for publication in Royal Society Open Science. The chemistry content of Royal Society Open Science is published in collaboration with the Royal Society of Chemistry.

On behalf of the Subject Editor Professor Anthony Stace and the Associate Editor Professor John Moses.

RSC Associate Editor
Comments to the Author:
(There are no comments.)

Reviewer(s)' Comments to Author:

Appendix A

Response to Reviewer comments to Author:

Publishing Editor:

Please also include the following statements alongside the other end statements. As we cannot publish your manuscript without these end statements included, if you feel that a given heading is not relevant to your paper, please nevertheless include the heading and explicitly state that it is not relevant to your work. We have included a screenshot example of the end statements for reference.

Ethical statement as well as Acknowledgement have been added (although our work falls into the field of pure synthetic organic chemistry, so it is obvious that no ethical assessment was required before before conducting the research).

> Reviewer: 1

>

> Comments to the Author(s)

> RSOS-181322

>

> This paper describes total synthesis of [15N]-labelled C6-substituted purines from [15N]-formamide - easy preparation of isotopically-labelled cytokinins and derivatives. Significant efforts have been devoted to the project, and some new chemistry data have been reported. The results are reasonable and support the conclusions. It fits well in the scope of RSOS, and it will be suitable for publication after careful revision.

>

> Comments:

> 1. References section is listed in alphabetical order, thus the references cited in the text is in a very unusual format, not starting from 1. Is this format RSOS requires?

All references have been renumbered as requested.

> 2. Too many references are cited in this manuscript, it might decrease the novelty of the work, making it an improvement of the synthetic approach.

Number of the references have been substantially reduced as requested.

> 3. References 71 and 72 seem not in alphabetical order. Please double check.

Renumbered as requested.

> 4. For solid compounds, it is better to give mp data.

Reviewer is right, mp data are usually given for solid compounds. On the other hand, it is not so common to measure mp for labelled compounds, as the method is destructive. However, we added mp data for the compounds where bigger quantities are available, as well as comparison with literature data for unlabelled counterparts.

> 5. For ¹H NMR data, it is better to give the proton assignments.

Added as requested.

> 6. HRMS, it is noted that the calcd value is exactly same with found value for several compounds. Please double check.

We double checked the original HR MS files and for some compounds the difference between calculated and measured value is really 0 ppm.

> 7. Within 11 compounds prepared, clearly point out how many new compounds and known compounds in Results and Discussion section.

Added as requested.

> 8. Page 14, Scheme 2 caption, there are some format issues: Italic, Superscript, Subscript; and compound number should be Bold.

Corrected as requested.

>

> Reviewer: 2

>

> Comments to the Author(s)

> The paper describes a practical and efficient synthesis of a useful labelled building block that should find use in the isotope labelling community.